# Variation in mental health-related sickness absence duration: The role of occupational health professionals

Sheila Timp[1]*, Nicky van Foreest[2], Willem van Rhenen[1,3]

1 Arbo Unie, Kennisinstituut Werk en Gezondheid, Nieuwegein, the Netherlands, 2 Faculty of Economics and Business, University of Groningen, Groningen, the Netherlands, 3 Center for Strategy, Organization and Leadership, Nyenrode Business Universiteit, Breukelen, the Netherlands

* sheila.timp@arbounie.nl

## Abstract

Mental health-related disorders are prevalent among the working population and account for a large proportion of sickness absence. Occupational health professionals (OHPs) play a key role in supporting employee recovery and reintegration, yet little is known about the extent to which individual OHPs affect return-to-work outcomes. Previous studies suggest that OHPs might influence absence duration, but comprehensive quantitative analyses examining variation between OHPs are lacking. In this study, we determine the variation in sickness absence duration attributable to OHPs in mental health-related cases. We analyze a large dataset of more than 30,000 sickness absence cases involving employees diagnosed with mental health-related disorders across multiple sectors. A cross-classified multilevel model was applied to estimate variance attributable to OHPs while controlling for other sources of variation. Our results show substantial variation in mental health-related sickness absence duration between OHPs ($\sigma^2$ = 0.13). This corresponds to marked differences in return-to-work outcomes: for high-performing OHPs, the median absence duration is 18 weeks or less, whereas for low-performing OHPs, the median duration is 28 weeks or more. Individual, organizational, and sectoral factors also contribute considerably to differences in sickness absence duration. Including OHP- and organization-level random effects reduced the estimated individual-level variance by about 50%, indicating that differences across providers and employers are relevant in reducing sickness absence duration. Based on this, we argue that OHPs and organizations should work in concert to reduce sickness absence durations. Future research should identify which specific OHP characteristics and practices are most helpful in accomplishing this.

**Data availability statement:** The data contain confidential medical and employment information and cannot be shared publicly under the terms of our ethics approval and data-use agreements. De-identified individual-level data may be available under controlled access from the Arbo Unie Data Team (datateam@arbounie. nl) for researchers who obtain ethics approval, sign a data-use agreement, and complete an approved Data Protection Impact Assessment (DPIA).

**Funding:** The author(s) received no specific funding for this work.

**Competing interests:** The authors have declared that no competing interests exist.

# 1 Introduction

Mental disorders are prevalent among the working population and significantly contribute to long-term sickness absence (LTSA), resulting in substantial costs for affected individuals, companies, and the national economy. The most common mental disorders associated with LTSA are depression, anxiety, and stress-related disorders such as distress, adjustment disorder, and burnout.

To better understand the impact of these disorders on sickness absence, several studies have investigated the duration and return-to-work rates for different mental health conditions [1–3]. These studies show that variation in sickness absence duration between sick-listed employees is substantial. This variation can result from factors related to the employee, their job or sector, the occupational health provider (OHP), the organization, or national systems. We refer to these as different levels of influence. In this study, we focus on variation attributable to OHPs while controlling for potential confounding factors at the employee, organizational, and sectoral levels.

Previous research primarily focused on variation at the employee and job levels. At the employee level, research shows that certain demographic factors increase sickness absence rates, including female gender and higher age [4,5]. Individual characteristics such as coping strategies, resilience, and personality traits also play an important role in sickness absence [6,7].

At the job level, several studies identified work-related factors risk factors including high emotional, physical, and mental demands, low autonomy, unclear role clarity, and limited work variety [8,9]. Alba-Jurado et al. found that sickness absence patterns vary across occupations, with more elemental occupations experiencing more frequent but shorter absences, where physical demands are higher and job autonomy is low [10]. However, other studies found no association between mental-related sickness absence and several work characteristics such as workload, work pace, variety and autonomy [11].

At higher levels of analysis, organizational research has examined characteristics that affect all employees within an organization, including organizational size and type, employment conditions, leadership styles, management practices, and return-to-work policies [12–14]. Research shows that smaller organizations tend to have lower absence rates, potentially due to closer interpersonal relationships, and less bureaucratic structures [14], while larger organizations often experience higher absence rates, possibly reflecting reduced social cohesion and increased anonymity [12].

At the sectoral level, research has documented differences in absence rates between employment sectors. For instance, healthcare and education show higher rates of mental health-related absences compared to other sectors, and systematic differences exist between the public and private sector [15–17].

Finally, at the national level, studies have found significant variation in sickness absence duration between countries, which can be attributed to differences in sickness absence regulations [12,18]. For instance, countries with generous sickness benefit systems, such as the Nordic countries and the Netherlands, consistently report longer absence periods.

The different levels of analysis are not clearly separable, as they often overlap and interact. For instance, certain sectors may have specific working conditions, such as high emotional labor demands in healthcare or teaching, making it difficult to distinguish between sectoral and job-level effects. Mastekaasa found that sector differences in sickness absence are largely explained by differences in employee characteristics (socio-demographic and individual factors like motivation) [16]. Moreover, factors may interact across levels. For example, Heinonen et al. report that the effect of job control on sickness absence depends on the sector [17]. Low job control was positively associated with mental health-related sickness absence in the health and social care sector and in the education sector, but in other sectors it was negatively associated.

Understanding how such multilevel factors interact has led to the development of comprehensive theoretical models. The effort-reward imbalance model by Siegrist proposes that an imbalance between high efforts and low rewards creates chronic stress [19]. Similarly, the job demands-resources model by Bakker and Demerouti suggests that employee well-being depends on the balance between job demands and available resources [20]. According to these models, problems arise when job demands exceed available resources (both job-related and personal), leading to exhaustion, stress-related disorders, and ultimately sickness absence. These theoretical frameworks emphasize that sickness absence results from complex interactions between individual, job, organizational, and societal factors rather than single-level determinants.

Despite the growing recognition of multilevel influences on sickness absence, the role of occupational health professionals (OHPs) in influencing sickness absence outcomes remains largely unexplored. Understanding variation between OHPs is particularly important given their central role in guiding employee recovery and return-to-work processes. Only a few studies, which we describe in the next section, have focused on the role of the OHP in sickness absence. Most of these studies are small experimental studies that compare guideline-trained OHPs with those providing care as usual, and do not use the multilevel structure that is present in occupational health practice.

To our knowledge, no large-scale studies have used multilevel modeling to examine variation in sickness absence duration between OHPs while controlling for other levels of influence. In this study, we fill this gap by quantifying how much individual OHPs contribute to variation in mental health-related sickness absence duration while controlling for employee, organizational, and sectoral factors. For this purpose, we analyze a large dataset from a Dutch occupational health service comprising employees from multiple sectors. In our dataset, each absence case is managed by the same OHP throughout the entire absence period. Since OHPs work for multiple organizations and organizations can be served by multiple OHPs, we use a cross-classified multilevel model (CCMM) to accurately estimate the effects of OHPs while controlling for individual, organizational and sectoral confounders. If substantial variation exists at the OHP level after controlling for other sources of variation, this would suggest that OHP practices represent an important target for intervention, alongside efforts at other levels. Such OHP-focused interventions could focus on improving communication skills, development of standardized protocols for return-to-work guidance, or implementation of peer learning programs where OHPs share best practices.

The paper is structured as follows. We begin with a review of the literature on variation between physicians in both general and occupational health care. We then describe the data and introduce the model, followed by the presentation of results. We conclude with a discussion of the findings and suggestions for future research.

## 2 Literature

Several studies in general health care have examined variation between physicians and/or hospitals. For instance, Maserejian et al. found that less experienced physicians adhered more closely to guidelines when managing musculo-skeletal conditions, suggesting that training may benefit more experienced clinicians [21]. Other studies report substantial variation in how primary care physicians manage common clinical scenarios [22,23]. Salet et al. analyzed differences in length of hospital stay, mortality, and healthcare costs, and found considerable variation between hospitals, while variation between individual physicians was small [24].

To estimate variation at multiple levels simultaneously, cross-classified multilevel models (CCMMs) can be used. These models account for complex data structures in which individuals are nested within more than one higher-level unit, such as neighborhoods and hospitals. Although the use of CCMMs in health care remains limited, it is increasing. Barker et al. identified 118 studies applying CCMM in health research [25], with common outcomes including body weight, subjective well-being, and substance use. Some of these studies use CCMM to estimate variation across both healthcare professionals and institutions. For example, Cafri and Fan applied CCMM to analyze hip implant survival, identifying considerable variation in revision rates attributable to both surgeons and hospitals [26]. Similarly, Pruitt et al. examined colorectal cancer screening and identified substantial variation across physicians, clinics, and neighborhoods [27]. Di Martino et al. used CCMM to study adherence to therapy for patients with cardiopulmonary diseases and found that hospitals contributed more to variation in adherence than primary care providers [28,29]. Overall, studies to sources of variation in healthcare show mixed results: some report greater variation between physicians, while others find that most variation occurs at the hospital level. The reasons for these inconsistencies require further investigation.

In occupational health, less is known about the extent to which occupational health professionals (OHPs) contribute to variation in recovery and return-to-work outcomes. Some prior studies in occupational health care focused on differences between OHPs [30–33]. Nieuwenhuijsen et al. (2003) evaluated the quality of occupational rehabilitation for employees with adjustment disorders and identified substantial room for improvement in performance indicators such as symptom assessment and continuity of care [30]. Van der Klink et al. (2003) demonstrated that training OHPs in time-contingent reintegration and cognitive-behavioral approaches significantly reduced long-term sickness absence [31]. Based on these findings, guidelines were developed to support employees with mental health-related disorders. Other studies have investigated whether adherence to these occupational health guidelines affects sickness absence duration. For example, Beurden et al. (2017) examined the relationship between guideline adherence and return-to-work duration for employees with mental disorders. They found that overall adherence to guidelines was low, limiting the analysis of its impact on absence duration. However, they observed that regular contact between the OHP and the employer was associated with earlier return-to-work, even when guideline adherence was minimal [33]. Similarly, Rebergen et al. (2009) investigated the impact of guideline-based care on sickness absence duration among employees with mental health-related disorders. While no effect was found for the entire group, they identified a small reduction in absence duration for employees with minor stress-related symptoms [32].

Most of these studies are small experimental studies in which one group received guideline training while the other provided care as usual. Beyond these studies, the influence of occupational health physicians (OHPs) on recovery and return-to-work outcomes remains largely unexplored.

## 3 Methods

### 3.1 Dataset

In the Netherlands, employees on sick leave have access to occupational health care, typically provided by an occupational health service (OHS). For the first two years of sick leave, employers are legally required to pay sick employees at least 70% of their most recent salary. Employees with short-term absences are generally not seen by an OHP. However, for longer absences, consultation with an OHP is mandatory before 42 days of absence. The OHP is responsible for medical guidance during the sickness absence.

For this study, we use data from a sickness absence register maintained by a large Dutch national OHS. The data were accessed for research purposes on 9 February 2025. Authors did not have access to directly identifying personal information during or after data collection. The Institutional Review Board of the Faculty of Economics and Business at the University of Groningen concluded that ethical clearance was not necessary for this study because the Medical Research involving Human Subjects Act does not apply to studies of anonymized register data. Therefore informed consent to participate and consent for publication were not applicable.

We include all reported sickness absence cases from January 2014 to December 2018 in which an OHP diagnosed a mental health disorder, such as adjustment disorder, burnout, depression, or anxiety. This time period was chosen so that most absence cases would conclude before the start of the COVID-19 pandemic. Cases are included if the following criteria are met: the employee is between 18 and 65 years old, the sickness absence duration is between one day and three years, and the employee was guided by the same OHP throughout the entire absence period (see Fig 1). The upper limit of three years reflects that some employers extend financial coverage beyond the mandatory two-year period, either because return to work is expected or due to more generous sickness absence regulations. For our analysis, we truncated absences lasting between two and three years to two years, since employers are only legally required to cover the first two years of sickness absence.

For the statistical analysis, we record the following variables for each absence case: the employee's age, gender, and diagnosis, the starting year of the sickness absence period, the occupational health professional (OHP) responsible for guiding the employee, and the organization and sector where the employee is employed. Mental health diagnoses were classified according to the Dutch classification system for Occupational and Social Insurance Physicians (CAS) [34], which is based on the International Statistical Classification of Diseases and Related Health Problems (ICD-10) [35]. We use the main mental diagnostic categories within the CAS system: distress (P1), adjustment disorders including burnout (P61), trauma-related disorders such as PTSD (P62), anxiety disorders (P63), personality disorders (P64), mood disorders including depression (P65), substance-related disorders (P66), and other mental health-related diagnoses.

The sector represents the working sector as specified in the Dutch Standard Industrial Classification (SBI) [36], that follows the International Standard Industrial Classification of All Economic Activities (ISIC). Our dataset includes 19 sectors, with education, healthcare, public administration, retail trade, and manufacturing being the most frequent. Table 1 presents each sector and the corresponding number of organizations and sickness cases. Each sector contains multiple organizations. An organization is defined as a company or institution that has a contract with the occupational health service (OHS), where all employees receive occupational health care from the same OHS. In our dataset, organizations vary considerably in size, with most having at least one thousand employees. The exact number of employees in each organization is unknown to the OHS due to Dutch privacy regulations. Our OHS focuses on clients from the public sector and large corporate organizations, such as hospitals, factories, municipalities, telecommunications companies, and retail chains.

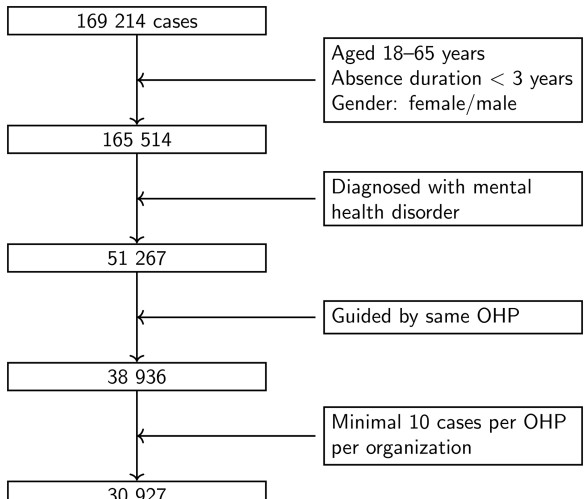

**Fig 1. Flowchart showing the inclusion of cases used in the analysis.**

**Table 1. Number of organizations and sickness absence cases by ISIC sector in the dataset.**

| ISIC | sector | # orgs | # cases |
|------|--------|--------|---------|
| A | Agriculture, forestry and fishing | 1 | 19 |
| B | Mining and quarrying | 2 | 22 |
| C | Manufacturing | 69 | 2,363 |
| D | Electricity, gas, steam and air conditioning supply | 3 | 905 |
| E | Water supply; sewerage, waste management and remediation | 9 | 341 |
| F | Construction | 16 | 1,020 |
| G | Wholesale and retail trade; repair of motor vehicles | 35 | 4,004 |
| H | Transportation and storage | 19 | 673 |
| I | Accommodation and food service activities | 1 | 72 |
| J | Information and communication | 6 | 709 |
| K | Financial and insurance activities | 23 | 570 |
| L | Real estate activities | 20 | 533 |
| M | Professional, scientific and technical activities | 35 | 1,796 |
| N | Administrative and support service activities | 18 | 686 |
| O | Public administration and defense; compulsory social security | 83 | 4,379 |
| P | Education | 121 | 6,826 |
| Q | Human health and social work activities | 45 | 5,190 |
| R | Arts, entertainment and recreation | 12 | 205 |
| S | Other service activities | 5 | 316 |
|  | Unspecified | 5 | 298 |
| **Total** | | **528** | **30,927** |

To obtain reliable estimates in multilevel analysis, it is important that group sizes are not too small [37]. Therefore, we restrict our dataset to absence cases guided by OHPs that have seen at least 10 different cases from the same organization.

### 3.2 Cross-classified model

Since we are interested in the duration of sickness absence until return-to-work, we are dealing with time-to-event data. The event is the moment an employee returns to work, which we define as recovery. To analyze such data, the Cox proportional hazard model $h_i(t) = h_0(t) \exp(X_i \beta)$ is commonly used, where $h_0(t)$ is the baseline recovery (hazard) function, and $X_i$ represents covariates for observation $i$.

In our study, OHPs can work for multiple organizations, and organizations can engage multiple OHPs, creating a cross-classified structure. Although frailty or mixed-effects Cox models can in principle accommodate multilevel data, current software provides limited and computationally demanding options for cross-classified designs [38]. Therefore, we model the survival process as a piece-wise exponential process that we can fit using a generalized linear model.

This approach accommodates multiple sources of random variation (e.g., OHP and organization effects) in cross-classified data and is well supported by generalized linear (mixed) models (GLMM) software, for background see [38–41].

We partition the absence duration of 104 weeks into 20 non-overlapping intervals, and determine cut points $0 = t_0 < t_1 < \ldots t_{20} = 104$, such that the number of recovery events is the same in each interval. We choose 20 intervals because the data show that the recovery rate is reasonably constant within each interval. Consequently, we model the baseline recovery rate within the $k$-th interval as the constant $h_k$.

To model variation at both the OHP and organizational levels, we use a cross-classified random effects structure, where each absence case is simultaneously linked to an OHP and to an organization. We estimate the recovery for each absence case $c$ in interval $k$ as:

$$h_{ck} = h_k \exp(X_c\beta + u_{m(c)} + v_{n(c)} + w_{i(c)}),$$

(1)

where each absence case $c$ corresponds to a specific absence period for the individual employee $i(c)$. For that case, $X_c$ denotes the fixed covariates (age, gender, diagnosis group, year the absence started, and sector). The term $m(c)$ refers to the organization (employer), and $n(c)$ to the treating OHP to whom individual $i(c)$ is linked during absence period $c$ (recall from Fig 1 that we included only cases guided by the same OHP throughout the entire absence period). We model organizational effects as $u_{m(c)} \sim N(0, \sigma_u^2)$, representing differences between organizations. OHP effects are specified as $v_{n(c)} \sim N(0, \sigma_v^2)$, capturing variation across providers. Individual effects are given by $w_{i(c)} \sim N(0, \sigma_w^2)$, accounting for person-specific unobserved factors and correlation across recurrent absence episodes from the same individual.

### 3.3 Model comparisons and experiments

To estimate the fixed and random effects, we construct from Eq. (1) four models with increasing complexity.

1. **Model A: Baseline Model** In this model, we simply neglect the random effects of the employee, the OHP, and the organization, so that Eq. (1) reduces to $h_{ck} = h_k \exp(X_c\beta)$. In this case, we can estimate $\beta$ and $\alpha_k = \log h_k$ by solving the linear model $\log \mu_{ck} = X_c\beta + \alpha_k + \log \gamma_{ck}$, where $\mu_{ck}$ is the estimated probability of recovery for absence case $c$ in period $k$, and $\gamma_{ck}$ is the exposure time for absence case $c$ in period $k$. Specifically, $\mu_{ck}$ is the total number of cases recovered in period $k$ divided by the number of sick cases 'entering' period $k$. The exposure time $\gamma_{ck} = t_k - t_{k-1}$ if the employee in absence case $c$ remains sick during the entire interval $k$, and $\gamma_{ck} = \tau_c - t_{k-1}$ if the recovery time $\tau_c$ for absence case $c$ occurs within the interval, i.e., $\tau_c \in [t_{k-1}, t_k)$.

2. **Model B: Add individual random effects**: In model B we add the random effect $w_{i(c)}$ of individual $i(c)$ to model A such that $h_{ck} = h_k \exp(X_c\beta + w_{i(c)})$

3. **Model C: Add OHP random effects**: In model C we add the random effect $v_{n(c)}$ of the OHP $n(c)$ that handled absence case $c$ to model B such that $h_{ck} = h_k \exp(X_c\beta + v_{n(c)} + w_{i(c)})$

4. **Model D: Add organizational random effects**: Finally, model D is the full model that includes all random effects for case $c$, i.e., for the individual $i(c)$, the organization $m(c)$ and the OHP $n(c)$, thereby resulting in Eq. (1).

We use the likelihood ratio test to compare nested models, and evaluate all models using the Akaike Information Criterion (AIC) and the Bayesian Information Criterion (BIC). To provide a practical interpretation of the OHP variance $\sigma_v^2$, Figs 4 and 3 illustrate the influence of these differences on the recovery rate and the survival curve.

Data pre-processing is performed in Python (version 3.12). For all four models, we use R (version 4.4.2) with the packages `survival` (version 3.8.3), `lme4` (version 1.1.35.5) and `ggplot2` (version 3.5.1). Model A (baseline model with fixed effects only) is fitted using the `glm()` function, while Models B-D (including random effects) are fitted using the `glmer()` function. The corresponding source code is provided in Supplementary Material S1 File (Code).

## 4 Results

### 4.1 Descriptive statistics

After applying our inclusion criteria, see Fig 1, the dataset consists of 30,927 absence cases involving 528 unique organizations and 280 different OHPs. Among the organizations, 152 (29%) were supported by multiple OHPs, while 191 (68%) OHPs worked with multiple organizations. Table 2 shows the cross-classification structure, with a total of 892 unique

**Table 2. This cross-classification table shows the distribution of OHP-organization relationships. Rows represent the number of organizations each OHP works with; columns represent the number of OHPs each organization is linked to. For instance, the third row, second column shows that there are 28 OHP-organization pairs where the OHP is linked to 3 organizations and the organization is linked to 2 OHPs.**

| OHPs working with | Organizations working with: | | | | | | | | |
| --- | --- | --- | --- | --- | --- | --- | --- | --- | --- |
| | 1 OHP | 2 OHPs | 3 OHPs | 4 OHPs | 5 OHPs | 6 OHPs | 7 OHPs | 8+OHPs | Total |
| 1 organization | 14 | 21 | 12 | 1 | 10 | 4 | 6 | 21 | 89 |
| 2 organizations | 24 | 16 | 16 | 3 | 9 | 10 | 6 | 14 | 98 |
| 3 organizations | 46 | 28 | 25 | 2 | 14 | 8 | 4 | 20 | 147 |
| 4 organizations | 57 | 24 | 10 | 6 | 4 | 4 | 2 | 13 | 120 |
| 5 organizations | 55 | 17 | 12 | 1 | 4 | 0 | 3 | 8 | 100 |
| 6 organizations | 42 | 18 | 6 | 4 | 6 | 3 | 2 | 9 | 90 |
| 7 organizations | 9 | 11 | 6 | 1 | 0 | 2 | 0 | 6 | 35 |
| 8 organizations | 33 | 8 | 5 | 1 | 4 | 2 | 1 | 2 | 56 |
| 9 organizations | 54 | 13 | 3 | 4 | 4 | 3 | 2 | 7 | 90 |
| 10 organizations | 11 | 6 | 2 | 0 | 0 | 0 | 0 | 1 | 20 |
| 11+organizations | 31 | 10 | 2 | 1 | 0 | 0 | 2 | 1 | 47 |
| Total | 376 | 172 | 99 | 24 | 55 | 36 | 28 | 102 | 892 |

OHP-organization relationships. There were only 14 OHP-organization pairs where the OHP works with only one organization and that organization works with only one OHP.

Table 3 presents statistics for our dataset, categorized by the year in which the absence began. The median absence duration is 24 weeks, ranging from 22 weeks in 2014–26 weeks in 2017. The average age across all year groups is approximately 44 years. Across all year groups, approximately 40% of employees are male. This over-representation of female employees can be explained by the high proportion of employees from the education and healthcare sectors in our dataset (see Table 1), which are sectors known for female predominance.

## 4.2 Exploratory analysis of absence duration

Fig 2 presents boxplots illustrating the distribution of median absence duration per OHP and per organization, respectively. Per box, the central black dot represents the median absence duration, whereas the height of the box spans the interquartile range (IQR), covering the middle 50% of the data. The boxplots are ordered according to increasing median absence duration. The horizontal red line indicates the population median absence duration, facilitating a comparative interpretation of variations in absence durations across different OHPs and organizations.

Fig 2 shows that the median absence duration per OHP ranges from 3 to 51 weeks, with 80% of OHPs having a median absence duration between 13 and 33 weeks. For organizations, the median absence duration spans 12–104 weeks, while 80% of organizations have a median absence duration between 14 and 36 weeks. The organizational

**Table 3. Basic information (mean and median absence duration in weeks, age, and gender) about the dataset, categorized by the year in which the absence began.**

| year | count | mean duration | median duration | age | % male |
| --- | --- | --- | --- | --- | --- |
| all | 30,927 | 31.0±25.3 | 23.9 | 44.3±11.2 | 39.3 |
| 2014 | 5940 | 28.5±24.4 | 21.6 | 44.3±10.9 | 40.2 |
| 2015 | 5997 | 30.5±25.5 | 23.4 | 44.4±11.0 | 40.4 |
| 2016 | 6290 | 30.5±25.5 | 23.0 | 44.4±11.3 | 40.3 |
| 2017 | 6506 | 33.1±25.9 | 25.9 | 44.5±11.5 | 37.6 |
| 2018 | 6194 | 32.1±25.2 | 25.1 | 44.0±11.5 | 38.1 |

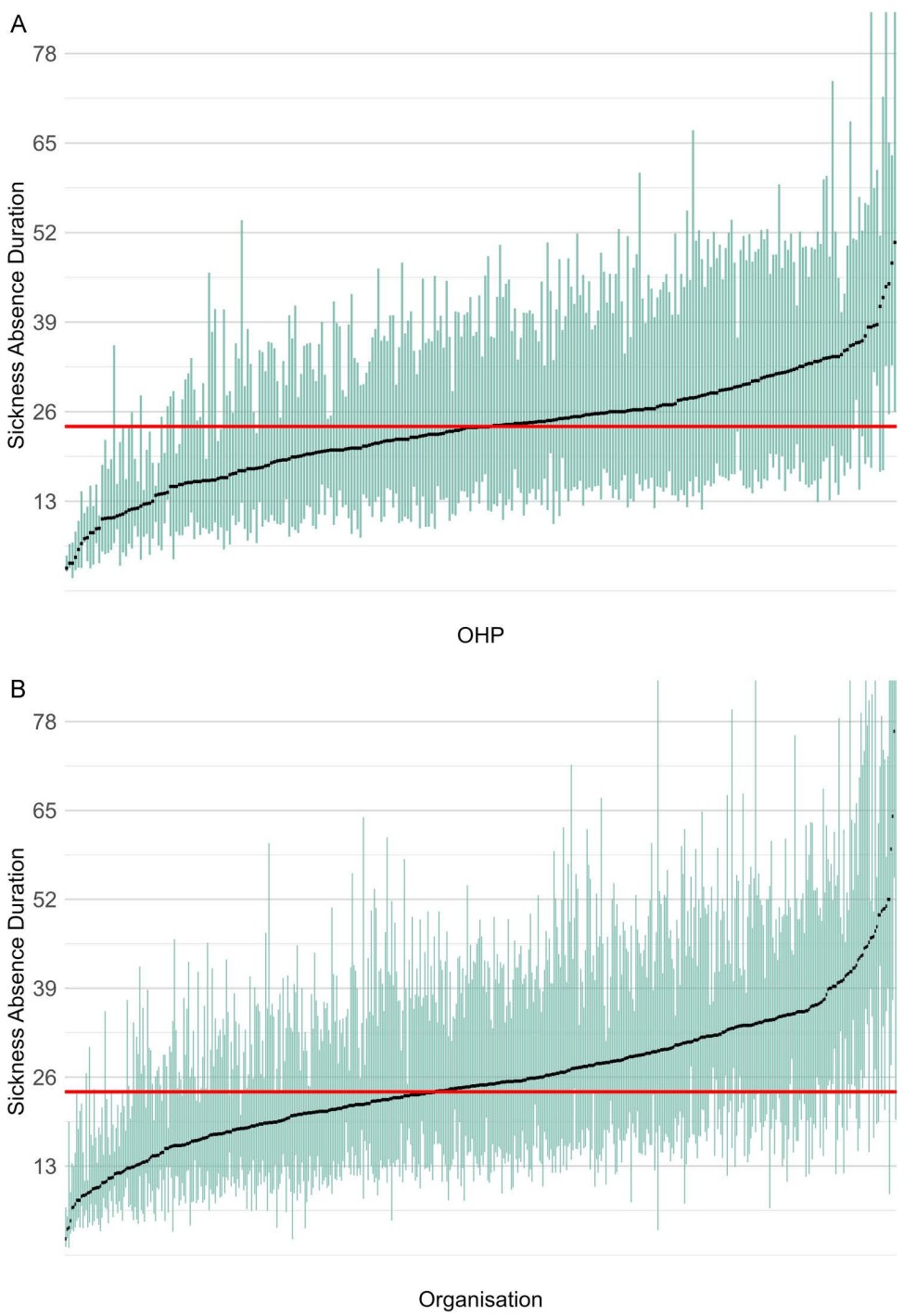

**Fig 2. The upper panel shows a boxplot illustrating the distribution of absence duration across 280 OHPs, the lower panel shows the boxplot for 528 organizations.** In each boxplot, the height of the box represents the interquartile range (IQR), covering the middle 50% of all absence durations for a specific OHP (organization). The black dot inside each box denotes the median absence duration for that OHP (organization). The horizontal red line indicates the median absence duration of the entire population.

distribution includes four notable outliers with a median absence duration more than 52 weeks, which might indicate problematic organizational characteristics.

The variation in absence duration is substantial for both OHPs and organizations. For a subset of OHPs and organizations, the interquartile range (IQR) does not include the population median, indicating that these entities perform significantly better or worse than average. The graphs suggest that organizational variation is similar to that of OHPs, with organizations showing more extreme outliers. However, the plots do not display the cross-classified effects of OHPs and organizations, which are presented in the model results in the next paragraph.

### 4.3 Model results

Table 4 presents the estimated parameters for the four models described in section 3.2. The coefficients for each time interval represent the logarithm of the baseline recovery rate for that interval. Across all models, the effect of gender is approximately 0.15, indicating that male employees have a higher likelihood of recovery than female employees. The age effect is approximately −0.004 per year, meaning that recovery likelihood slightly decreases with each additional year of age. With respect to diagnosis, distress (the reference category) is associated with the shortest absence durations, whereas personality disorders are associated with the longest.

For sector effects, we use the education sector as the reference category since it is the largest sector in our dataset. All other sectors show positive coefficients, indicating that the education sector has the slowest recovery rate. The sector with the fastest recovery is accommodation and food service, where recovery is exp(0.86) = 2.4 times faster than in education.

Model A includes only fixed effects, while Model B adds an individual-level random effect. In Model B, the estimated variance of the individual random effect is relatively large at 0.45. The relatively large variance indicates substantial individual differences in recovery rates even after controlling for observed characteristics like age, gender, diagnosis, and sector.

Model C extends Model B by including an OHP random effect with estimated variance of 0.18. The estimated variance of the individual decreases from $\sigma_w^2$ = 0.44 to $\sigma_w^2$ = 0.27, suggesting that part of the variance that was attributed to individual differences can be explained by differences between OHPs. The likelihood ratio test shows that Model C provides a significantly better fit than Model B ($\chi^2(1)$ = 1031, $p \ll 0.001$).

In the cross-classified random effects model (Model D), the estimated variance components are $\sigma_u^2$ = 0.10 for the organization, and $\sigma_v^2$ = 0.13 for the OHP, indicating that the OHP-level random effect contributes slightly more to the overall variation than the organization-level effect. Likelihood ratio tests confirm that the cross-classified model significantly improves model fit over both Model B ($\chi^2(2)$ = 1484, $p \ll 0.001$) and Model C (($\chi^2(1)$ = 452, $p \ll 0.001$). The cross-classified model shows that adding the organization random effect influences the OHP random effect (the variance decreases from 0.18 to 0.13), suggesting that part of what was initially attributed to OHP differences is actually due to organization differences. The individual variance slightly decreases from $\sigma_w^2$ = 0.27 to $\sigma_w^2$ = 0.25. From this model we conclude that organizations and OHPs contribute as much to the variance as individuals.

In the next paragraph, we assess the practical implications of variation between OHPs on recovery rates and survival curves using the cross-classified model (see Figs 3 and 4).

### 4.4 Effect of OHP variance on recovery rate & survival curve

We determine recovery rates and return-to-work survival curves for an average employee seen by an average OHP, as well as for employees guided by OHPs whose random effect is one standard deviation above or below average. These analyses are based on the results of the model Model D (cross-classified model including OHP and organization random effects) and the findings are illustrated in Figs 3 and 4.

Fig 3 shows the variation in OHP-specific recovery rates. With a random effect variance of $\sigma_v^2$ = 0.13, the recovery rate for an employee guided by an OHP one standard deviation above the average increases by a factor of

**Table 4. This table shows the fixed and random coefficients for the four different models. Model A only includes fixed effects. Model B includes fixed effects and a random effect for the individual employee; model C includes fixed effects and random effects for both the individual and the OHP. Model D is the cross-classified model that adds the organization random effect to model C.**

| Model | Model A | Model B | Model C | Model D |
|---|---|---|---|---|
| | fixed effects | +individual | + OHP | + organization |
| male | 0.13±0.01 | 0.18±0.02 | 0.14±0.02 | 0.12±0.02 |
| age (gdc) | −0.004±0.001 | −0.005±0.001 | −0.004±0.001 | −0.004±0.001 |
| year 2014 (ref) | 0.00±0.00 | 0.00±0.00 | 0.00±0.00 | 0.00±0.00 |
| year 2015 | −0.08±0.02 | −0.10±0.02 | −0.04±0.02 | −0.01±0.02 |
| year 2016 | −0.06±0.02 | −0.08±0.02 | −0.03±0.02 | 0.01±0.02 |
| year 2017 | −0.16±0.02 | −0.22±0.02 | −0.12±0.02 | −0.06±0.02 |
| year 2018 | −0.11±0.02 | −0.17±0.02 | −0.06±0.02 | 0.02±0.02 |
| distress (P1) (ref) | 0.00±0.00 | 0.00±0.00 | 0.00±0.00 | 0.00±0.00 |
| adjustment disorder (P61) | −0.50±0.02 | −0.71±0.03 | −0.84±0.03 | −0.84±0.03 |
| PTSD/trauma related disorder (P62) | −0.77±0.04 | −1.01±0.05 | −1.15±0.05 | −1.15±0.05 |
| anxiety disorder (P63) | −0.92±0.05 | −1.21±0.06 | −1.29±0.06 | −1.29±0.06 |
| personality disorder (P64) | −1.16±0.10 | −1.42±0.12 | −1.52±0.11 | −1.48±0.11 |
| mood disorder (P65) | −1.04±0.03 | −1.39±0.04 | −1.49±0.04 | −1.49±0.04 |
| substance-related disorder (P66) | −0.85±0.11 | −1.08±0.13 | −1.27±0.12 | −1.21±0.13 |
| other | −0.54±0.02 | −0.71±0.03 | −0.77±0.03 | −0.76±0.03 |
| A Agriculture, forestry and fishing | 0.40±0.23 | 0.47±0.30 | 0.53±0.32 | 0.46±0.47 |
| B Mining and quarrying | 0.28±0.22 | 0.39±0.28 | 0.49±0.27 | 0.43±0.37 |
| C Manufacturing | 0.17±0.02 | 0.24±0.03 | 0.22±0.04 | 0.27±0.07 |
| D Electricity, gas, steam and air conditioning | 0.18±0.04 | 0.22±0.05 | 0.40±0.06 | 0.24±0.22 |
| E Water supply; sewerage, waste management | 0.01±0.06 | −0.01±0.07 | 0.08±0.08 | 0.07±0.14 |
| F Construction | 0.18±0.03 | 0.26±0.05 | 0.14±0.05 | 0.17±0.11 |
| G Wholesale and retail trade | 0.32±0.02 | 0.42±0.03 | 0.35±0.04 | 0.34±0.08 |
| H Transportation and storage | 0.38±0.04 | 0.47±0.05 | 0.41±0.07 | 0.35±0.11 |
| I Accommodation and food service | 0.86±0.12 | 0.99±0.15 | 0.82±0.18 | 0.82±0.39 |
| J Information and communication | 0.63±0.04 | 0.87±0.05 | 0.41±0.07 | 0.35±0.16 |
| K Financial and insurance activities | 0.16±0.04 | 0.20±0.06 | 0.15±0.06 | 0.31±0.10 |
| L Real estate activities | 0.24±0.05 | 0.30±0.06 | 0.33±0.06 | 0.32±0.10 |
| M Professional, scientific and technical | 0.14±0.03 | 0.15±0.03 | 0.15±0.04 | 0.06±0.08 |
| N Administrative and support service | 0.09±0.04 | 0.10±0.05 | 0.13±0.06 | 0.19±0.11 |
| O Public administration and defense | 0.20±0.02 | 0.25±0.03 | 0.26±0.03 | 0.22±0.06 |
| P Education (ref) | 0.00±0.00 | 0.00±0.00 | 0.00±0.00 | 0.00±0.00 |
| Q Human health and social work | 0.11±0.02 | 0.18±0.02 | 0.13±0.03 | 0.12±0.07 |
| R Arts, entertainment and recreation | 0.26±0.07 | 0.32±0.09 | 0.25±0.09 | 0.26±0.14 |
| S Other service activities | 0.12±0.06 | 0.13±0.07 | 0.12±0.08 | 0.11±0.19 |
| Unknown sector | 0.47±0.06 | 0.57±0.08 | 0.57±0.11 | 0.55±0.19 |
| var individual $\sigma_w^2$ | | 0.44 | 0.27 | 0.25 |
| var OHP $\sigma_v^2$ | | | 0.18 | 0.13 |
| var organization $\sigma_u^2$ | | | | 0.10 |
| AIC | 237872 | 237377 | 236342 | 235891 |
| BIC | 238444 | 237960 | 236936 | 236496 |
| logLik | −118883 | −118634 | −118116 | −117890 |

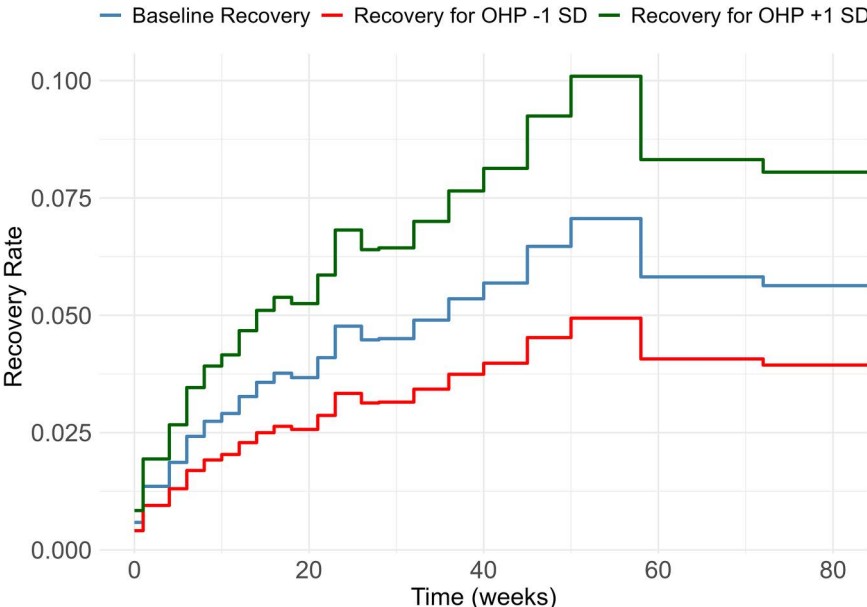

**Fig 3. Recovery rates for employees guided by an average OHP, and by OHPs performing one standard deviation below (red line) or above (green line) average.** The curves show that the weekly recovery rate is approximately twice as high for OHPs performing one standard deviation above average compared to those performing one standard deviation below average.

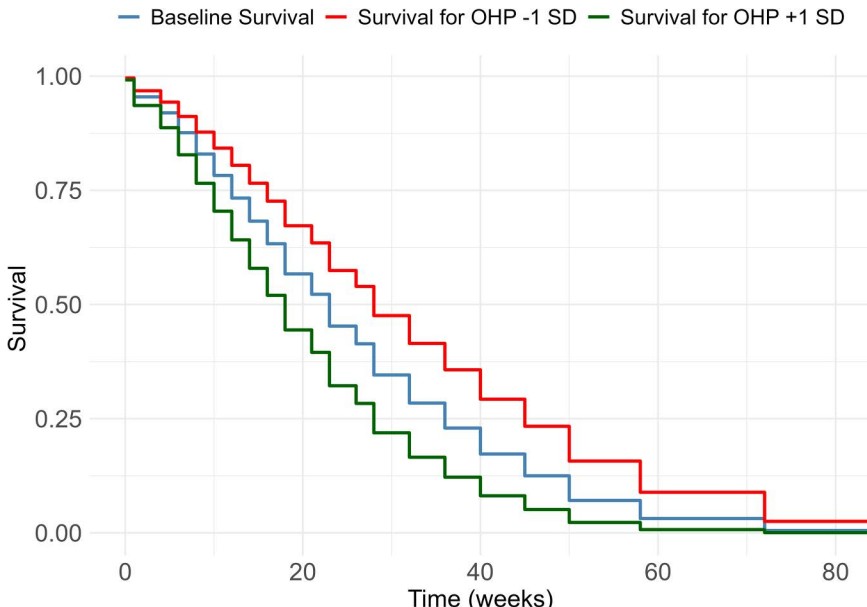

**Fig 4. Survival curve for employees guided by an average OHP, and by OHPs performing one standard deviation below (red line) or above (green line) average.** The curves show that the median duration for above-average OHPs is around 18 weeks, while for below-average OHPs, it is 28 weeks.

$\exp(\sigma_v) = \exp(0.36) = 1.43$, while for an employee guided by an OHP one standard deviation below the average, it decreases by a factor of $\exp(\sigma_v) = \exp(-0.36) = 0.70$. For example, at 40 weeks, the recovery rate is 0.057 for an average OHP, 0.039 for an OHP performing one standard deviation below average, and 0.081 for an OHP performing one standard deviation above average. This implies that, at this point in time, OHPs one standard deviation above average achieve approximately double the weekly recovery rate compared to OHPs one standard deviation below average. The absolute differences in recovery rates between OHPs increase up to approximately one year and then stabilize. Additionally, a noticeable peak in recovery rates appears around one year of absence, resulting from changes in sickness absence policies at that time [42].

Fig 4 illustrates the variation in OHP-specific return-to-work survival curves. During the first three months, differences between OHPs remain relatively small, with most employees still absent from work. By this point, return-to-work percentages for most OHPs range between 13% and 26%, with roughly one-third of OHPs performing either below or above this range.

As time progresses, these differences become more pronounced. After six months, the majority of OHPs have return-to-work percentages around 40%. However, for OHPs one standard deviation below average, only 28% of employees have returned, while for OHPs one standard deviation above average, about 54% have resumed work. After one year, the majority of OHPs have return-to-work percentages around 93%. Among OHPs one standard deviation below average, only 84% of employees have returned, whereas for those one standard deviation above average, around 98% have successfully returned to work.

From Fig 4 we can also compare OHPs based on the median absence duration. Overall, the median absence duration is approximately 23 weeks. For OHPs one standard deviation above average, the median duration is around 18 weeks, while for those one standard deviation below average, it is around 28 weeks. These results illustrate that, despite the relatively small estimated random effect variance, the impact of the OHP on employee recovery and return-to-work outcomes is substantial.

## 5 Discussion

This study investigates the variation in recovery rates (and the resulting sickness absence durations) attributable to differences between OHPs. Results show that substantial variation exists among OHPs, with considerable differences in absence duration ranging from approximately 18 weeks for OHPs one standard deviation above average to 28 weeks for OHPs one standard deviation below average. Above-average OHPs achieve approximately double the weekly recovery rate compared to below-average OHPs at key time points during the absence period.

All models show large differences in recovery rates between sectors, with the public sectors education and healthcare associated with the lowest recovery rates, aligning with prior studies [15,16]. Also consistent with existing literature, female gender and older age are associated with longer absence durations [4,5].

About half of the variance that was first attributed to individuals (Model B) could be explained by differences between OHPs and organizations (model D). Specifically, individual level random variation decreases from $\sigma_w^2 = 0.44$ to $\sigma_w^2 = 0.25$. After accounting for fixed individual characteristics (such as age and gender), the remaining unexplained individual variation is comparable to the variation of OHPs ($\sigma_v^2 = 0.13$) and organizations ($\sigma_u^2 = 0.10$) together.

Our findings show the critical role that OHPs and organizations play in employee recovery. Based on this, we argue that OHPs and organizations should work in concert to improve return-to-work outcomes. OHPs are well-positioned to take a leading role in this collaborative process, as they often consult across multiple organizations and can facilitate the transfer of best practices between them. Additionally, OHPs can enhance their effectiveness through peer learning and knowledge sharing with other professionals.

### 5.1 Strengths and limitations

A key strength of this study is the use of a large, comprehensive dataset comprising over 30,000 sickness absence cases from 280 OHPs across 528 organizations and 19 economic sectors. This large set enables robust estimation of variation

attributable to multiple levels simultaneously. In addition, the application of a cross-classified multilevel model accounts for the non-hierarchical structure where OHPs work for multiple organizations and organizations engage multiple OHPs. Previous studies have been limited by smaller sample sizes and simpler statistical approaches that could not disentangle these cross-classified effects. Another strength is the inclusion of multiple sectors, which strengthens the generalizability of the results.

However, this study also has some limitations. First, while we controlled for several variables (age, gender, diagnosis, organization, sector), unmeasured confounders at each level may still influence the results. For instance, we lack detailed information about specific employee characteristics (coping), job roles within organizations, or workplace psychosocial factors. These factors might influence and/or interact with factors at the OHP level. Second, the study focuses on mental health-related disorders, and findings may not generalize to other types of sickness absence such as musculoskeletal or cardiovascular conditions. Third, we have not identified which specific OHP characteristics drive the observed variation, as our data contain only the variation attributable to OHPs. In fact, we believe that our findings provide the motivation for a detailed study of differences in OHP practices. Finally, the data come from a single Dutch occupational health service, in which some sectors–like health and education–are over-represented.

### 5.2 Comparison with previous studies

To our knowledge, this is the first study to examine variation across OHPs while controlling for other levels of variation using a cross-classified multilevel model (CCMM). Some prior studies in occupational health care have studied the effect of guideline adherence on the performance of OHPs [31–33]. Initially, van der Klink et al. demonstrated that certain activating practices by occupational physicians, including cognitive behavioral techniques and work-focused interventions, were effective in reducing sickness absence duration [31]. These findings provided the evidence base for developing the Dutch occupational mental health guideline [43].

However, studies evaluating the implementation and effectiveness of these guidelines have reported limited effects. For example, Rebergen et al. found in a study of 240 police employees that guideline-based care did not shorten sickness absence compared to usual care, though modest benefits were observed among administrative staff with minor stress-related symptoms [32]. Similarly, van Beurden et al. studied 114 workers and found low overall adherence to guideline recommendations by OHPs; this adherence was not associated with return-to-work outcomes, although specific elements such as regular employer contact showed potential [33]. Notably, these implementation studies involved relatively small samples of 100–200 employees, which may have limited their ability to detect effects.

In contrast, our study identifies substantial differences among OHPs in return-to-work outcomes. Rather than evaluating guideline adherence directly, we quantify the magnitude of between-OHP variation in sickness absence duration. This variation may reflect other factors such as OHP communication skills, clinical experience, or work style. At the same time, the mixed evidence from prior studies suggests that guideline adherence may influence absence duration and warrants investigation in large-scale studies.

In the general care setting, several studies have examined variation at multiple levels, such as the level of the physician and the hospital [22–24,26–29]. These studies report inconclusive results regarding whether physicians or hospitals contribute more to the observed variation, although most suggest a greater contribution from hospitals. In our study, we found that both OHP and organizational variation contribute substantially to variation in sickness absence outcomes, while the OHP effect ($\sigma_v^2$ = 0.13) is only slightly larger than the organizational effect ($\sigma_u^2$ = 0.10).

This finding differs from the pattern often observed in general healthcare, where hospital-level factors typically dominate over individual physician effects. Several factors may explain why OHP and organizational effects are more balanced in occupational health settings. First, by controlling for sector as a fixed effect in our models, we account for systematic differences between industries (e.g., education vs. manufacturing vs. healthcare), which removes a major source of between-organization variation that might otherwise be attributed to organizational effects. Second, all organizations in

our dataset operate under the same occupational health regulations, both at the national level and from the OHS provider, which may reduce organizational variance. Third, OHPs in occupational health may have more autonomous decision-making authority regarding return-to-work recommendations compared to physicians in hospital settings, where institutional protocols may be more constraining.

### 5.3 Implications and future research

As this study was purely quantitative, in-depth research is needed to identify which OHP- and organization-level factors contribute to the observed variation. Potential OHP-level factors include professional experience, communication style during consultations, the type and quality of advice provided to employees and employers, adherence to guidelines, and the frequency and quality of contact with employers and other professionals (e.g., medical specialists, psychologists, labour experts). At the organization level, factors such as HR policies, return-to-work procedures, case-management capacity, industry sector, and firm size may play a role.

Assessing guideline use has been resource-intensive in prior work–for example, Beurden et al. manually reviewed medical records to evaluate adherence [33]. Recent advances in natural-language processing, including large language models (LLMs), offer scalable ways to analyze medical records and OHP advice notes. These methods could be used to estimate guideline adherence and may also provide indirect indicators of communication style.

Insights from these analyses could inform targeted interventions to enhance OHP effectiveness and support peer-learning initiatives aimed at reducing sickness-absence duration, thereby facilitating employee reintegration and lowering the costs associated with prolonged absence.

## Supporting information

**S1 File. Source code.**

(R)

## Author contributions

**Conceptualization:** Sheila Timp, Willem van Rhenen.

**Data curation:** Sheila Timp.

**Formal analysis:** Sheila Timp, Nicky van Foreest.

**Methodology:** Sheila Timp, Nicky van Foreest.

**Software:** Sheila Timp.

**Writing – original draft:** Sheila Timp.

**Writing – review & editing:** Sheila Timp, Nicky van Foreest, Willem van Rhenen.

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
