## [Decision Letter · Decision Letter 0]

23 Jul 2025

PONE-D-25-30340Variation in Mental Health-Related Sickness Absence: The Role of Occupational Health Professionals and OrganizationsPLOS ONE

Dear Dr. Timp,

Thank you for submitting your manuscript to PLOS ONE. After careful consideration, we feel that it has merit but does not fully meet PLOS ONE’s publication criteria as it currently stands. Therefore, we invite you to submit a revised version of the manuscript that addresses the points raised during the review process.

**ACADEMIC EDITOR:**

I appreciate the opportunity to evaluate this manuscript, which addresses a relevant topic in occupational and public health. Both reviewers provided constructive and detailed feedback, and I kindly request that the authors revise the manuscript thoroughly, addressing all points below to ensure the work meets PLOS ONE's publication standards.

**Comments from Reviewer 1:**

1) The authors make conclusions that are not strongly supported by their data. In particular, they state that their “findings suggest that interventions aiming to improve sickness absence management should focus primarily on enhancing OHP practices rather than on organizational reforms.” However, they studied a very narrow slice of organizations and specifically tried to reduce the amount of variation between organizations. As stated by the authors “Furthermore, we minimized variation at other levels by selecting employees working in the same sector in the same country, which strengthens the reliability of the result.” Given that the authors explicitly sought to minimize variation at the organizational level in their analyses, it is not surprising that their results show greater variation at the provider level than at the organizational level. It is likely that variation at the level of the organization would be much larger if data were not restricted only to organizations in the educational sector. This design is useful to show that there is large variation in sickness absence at the level of the provider within one employment sector. However, concluding broadly that organizational level interventions should be ignored in favor of provider level interventions is a serious over-generalization of their findings.

2) The literature review is rather sparse and some of the citations were puzzling. For instance, citation #4 was cited in conjunction with this sentence “At the job level, several studies have investigated work-related factors and employer characteristics”. However, the goal of this paper was to determine “whether adherence to the Dutch occupational mental health guideline by occupational physicians was associated with time to return to work in workers sick-listed due to common mental disorders” – i.e. the paper focused on provider behavior not on employer organizational characteristics. Largely ignored by the authors is the rather large literature demonstrating variations in the incidence/prevalence of mental health issues between workers in different industries, and the role of organizational effects and organizational level interventions to promote mental health among working populations.

3) This paper focuses narrowly on the duration of sickness absence (what happens after you take time off from work for a mental health disorder). It would also be important to report if there was variation in the proportion of workers who took sickness absence between organizations, not just the duration of illness.

Given the complexity of return-to-work, this research would benefit from a robust theoretical framework to conceptualize the determinants of RTW, as well as better alignment between observational findings and policy recommendations. The study addresses an important question but requires substantial revision given the narrow slice of employers that were studied.

4) There are some analytic points that could be better explained. The authors provided complete and exhaustive descriptions of the variation in sick leave duration by provider and by organization. As noted above, they unfortunately restricted their analysis to one employment sector, limiting their ability to support their broader conclusions about the relative importance of organization level factors vs. provider factors.

5) Another ambiguity was noted. When an employee had multiple periods of absence, each period was included in the dataset as a single case. It is not clear to the reader if one individual could contribute multiple periods to the data (and if so, were these absences treated as independent or clustered events?)

6) The reader does not know what diagnoses or types of mental health disorders included in the cases analyzed. It is likely that different OHPs saw different mixes of cases, which might have different expected recovery times. Variation in OHP performance might be in part attributable to differences in the types of mental health conditions that they saw.

7) The authors' decision to include data from 2016 to December of 2020 data to avoid COVID-19 effects is questionable, as the pandemic's psychological and workplace impacts likely began influencing mental health and absence patterns well before the December 2020 cutoff for their study. Most people would not consider this to represent a "pre-pandemic" sample.

8) One point is unclear – the authors never defined what they meant by the term “organization.” This appears to mean the employers of the workers studied, but could also mean a trade union or other labor organization. It is not clear the type or size of the employers or organizations.

9) There are a few minor misspellings (cardiopulmonair, Februari).

10) This study addresses an important occupational health question using substantial data, but several fundamental issues prevent it from supporting its conclusions. As pointed out above, the finding of greater variation by provider than by organization is unsurprising, given the narrow bounding of organizations. A theoretical framework to explain findings would be helpful. A major revision is needed to align methodology with conclusions, before this manuscript can adequately inform occupational health practice. A deeper dive into the characteristics of OHPs that predicted variation in absence time would be interesting. The current conclusion that “interventions aiming to improve sickness absence management should focus primarily on enhancing OHP practices rather than on organizational reforms” is not supported by the data provided.

**Comments from Reviewer 2:**

1) Which model parameter estimation method was used: Maximum Likelihood or Monte Carlo?

2) Which function from the lme4 package was used? And what were the arguments adopted in this function?

3) I would like to see the cross-classification table to be able to assess the degree of cross-hierarchy.

4) The sentence "...However, applying this model requires that the data follow either a two-level hierarchical structure (with observations nested within a single clustering variable) or strictly nested clusters..." The Cox proportional hazard model does not require a hierarchical structure. This sentence seems inappropriate, or I didn't understand the meaning the authors intended.

**Academic Editor:**

In summary, while the manuscript addresses a relevant topic in occupational health and is based on a robust dataset, several critical issues raised by Reviewer 1 must be carefully addressed before the work can be reconsidered for publication. Most notably, the main conclusion - suggesting that interventions should focus primarily on enhancing OHP practices rather than on organizational reforms - does not appear to be sufficiently supported by the data presented. This inference appears overstated, particularly given that the study design deliberately restricted organizational-level variation by including only organizations within the education sector.

It seems to me that the greater variation observed at the OHP level is to be expected in this context and does not, in itself, justify prioritizing provider-focused interventions over organizational ones. Additionally, the manuscript does not investigate or describe specific practices or characteristics of OHPs that could explain the observed differences in outcomes, limiting the practical utility of this recommendation.

I would also like to reinforce the reviewer’s concern regarding the lack of clarity about what constitutes an “organization” in the analysis. Although organizations are a central comparative unit in the study, the manuscript does not provide a clear definition of what constitutes an “organization”. It appears to refer to the employer of the employee, likely individual schools or educational institutions, but this is not explicitly stated. Furthermore, no details are provided regarding the type (e.g., primary, secondary, higher education), size, or structural features of these organizations. This lack of specificity weakens the interpretation of “organizational-level effects” and limits the reader’s ability to assess the plausibility and relevance of the comparisons made.

Given these issues, I encourage the authors to revise the manuscript substantially, aligning the conclusions more closely with the scope and limitations of the data, clarifying key definitions, and addressing the methodological and conceptual concerns raised by the reviewers.

If applicable, we recommend that you deposit your laboratory protocols in protocols.io to enhance the reproducibility of your results. Protocols.io assigns your protocol its own identifier (DOI) so that it can be cited independently in the future. For instructions see: https://journals.plos.org/plosone/s/submission-guidelines#loc-laboratory-protocols. Additionally, PLOS ONE offers an option for publishing peer-reviewed Lab Protocol articles, which describe protocols hosted on protocols.io. Read more information on sharing protocols at . Additionally, PLOS ONE offers an option for publishing peer-reviewed Lab Protocol articles, which describe protocols hosted on protocols.io. Read more information on sharing protocols at https://plos.org/protocols?utm_medium=editorial-email&utm_source=authorletters&utm_campaign=protocols..

We look forward to receiving your revised manuscript.

Kind regards,

João Marcos Bernardes, Ph.D. in Public Health

Academic Editor

PLOS ONE

Journal Requirements:

2. In the online submission form, you indicated that [The data that support the findings of this study are available from the authors upon reasonable request.].

Reviewers' comments:

Reviewer's Responses to Questions

**Comments to the Author**

1. Is the manuscript technically sound, and do the data support the conclusions?

Reviewer #1: Partly

Reviewer #2: Yes

2. Has the statistical analysis been performed appropriately and rigorously? 

Reviewer #1: No

Reviewer #2: Yes

3. Have the authors made all data underlying the findings in their manuscript fully available?

Reviewer #1: Yes

Reviewer #2: No

4. Is the manuscript presented in an intelligible fashion and written in standard English?

Reviewer #1: Yes

Reviewer #2: Yes

5. Review Comments to the Author

Reviewer #1: Is the manuscript technically sound, and do the data support the conclusions?

The authors make conclusions that are not strongly supported by their data. In particular, they state that their “findings suggest that interventions aiming to improve sickness absence management should focus primarily on enhancing OHP practices rather than on organizational reforms.” However, they studied a very narrow slice of organizations and specifically tried to reduce the amount of variation between organizations. As stated by the authors “Furthermore, we minimized variation at other levels by selecting employees working in the same sector in the same country, which strengthens the reliability of the result.” Given that the authors explicitly sought to minimize variation at the organizational level in their analyses, it is not surprising that their results show greater variation at the provider level than at the organizational level. It is likely that variation at the level of the organization would be much larger if data were not restricted only to organizations in the educational sector. This design is useful to show that there is large variation in sickness absence at the level of the provider within one employment sector. However, concluding broadly that organizational level interventions should be ignored in favor of provider level interventions is a serious over-generalization of their findings.

The literature review is rather sparse and some of the citations were puzzling. For instance, citation #4 was cited in conjunction with this sentence “At the job level, several studies have investigated work-related factors and employer characteristics”. However, the goal of this paper was to determine “whether adherence to the Dutch occupational mental health guideline by occupational physicians was associated with time to return to work in workers sick-listed due to common mental disorders” – i.e. the paper focused on provider behavior not on employer organizational characteristics. Largely ignored by the authors is the rather large literature demonstrating variations in the incidence/prevalence of mental health issues between workers in different industries, and the role of organizational effects and organizational level interventions to promote mental health among working populations.

This paper focuses narrowly on the duration of sickness absence (what happens after you take time off from work for a mental health disorder). It would also be important to report if there was variation in the proportion of workers who took sickness absence between organizations, not just the duration of illness.

Given the complexity of return-to-work, this research would benefit from a robust theoretical framework to conceptualize the determinants of RTW, as well as better alignment between observational findings and policy recommendations. The study addresses an important question but requires substantial revision given the narrow slice of employers that were studied.

Has the statistical analysis been performed appropriately and rigorously?

There are some analytic points that could be better explained –

The authors provided complete and exhaustive descriptions of the variation in sick leave duration by provider and by organization. As noted above, they unfortunately restricted their analysis to one employment sector, limiting their ability to support their broader conclusions about the relative importance of organization level factors vs. provider factors.

Another ambiguity was noted. When an employee had multiple periods of absence, each period was included in the dataset as a single case. It is not clear to the reader if one individual could contribute multiple periods to the data (and if so, were these absences treated as independent or clustered events?)

The reader does not know what diagnoses or types of mental health disorders included in the cases analyzed. It is likely that different OHPs saw different mixes of cases, which might have different expected recovery times. Variation in OHP performance might be in part attributable to differences in the types of mental health conditions that they saw.

The authors' decision to include data from 2016 to December of 2020 data to avoid COVID-19 effects is questionable, as the pandemic's psychological and workplace impacts likely began influencing mental health and absence patterns well before the December 2020 cutoff for their study. Most people would not consider this to represent a "pre-pandemic" sample..

Is the manuscript presented in an intelligible fashion and written in standard English?

The manuscript is clear and well-written in standard English.

One point is unclear – the authors never defined what they meant by the term “organization.” This appears to mean the employers of the workers studied, but could also mean a trade union or other labor organization. It is not clear the type or size of the employers or organizations.

There are a few minor misspellings (cardiopulmonair, Februari).

Review Comments to the Author

This study addresses an important occupational health question using substantial data, but several fundamental issues prevent it from supporting its conclusions. As pointed out above, the finding of greater variation by provider than by organization is unsurprising, given the narrow bounding of organizations. A theoretical framework to explain findings would be helpful. A major revision is needed to align methodology with conclusions, before this manuscript can adequately inform occupational health practice. A deeper dive into the characteristics of OHPs that predicted variation in absence time would be interesting.

The current conclusion that “interventions aiming to improve sickness absence management should focus primarily on enhancing OHP practices rather than on organizational reforms” is not supported by the data provided.

Reviewer #2: 1- Which model parameter estimation method was used: Maximum Likelihood or Monte Carlo?

2- Which function from the lme4 package was used? And what were the arguments adopted in this function?

3- I would like to see the cross-classification table to be able to assess the degree of cross-hierarchy.

4- The sentence "...However, applying this model requires that the data follow either a two-level hierarchical structure (with observations nested within a single clustering variable) or strictly nested clusters..." The Cox proportional hazard model does not require a hierarchical structure. This sentence seems inappropriate, or I didn't understand the meaning the authors intended.

6. PLOS authors have the option to publish the peer review history of their article (what does this mean?). If published, this will include your full peer review and any attached files.). If published, this will include your full peer review and any attached files.

.

Reviewer #1: No

Reviewer #2: No

While revising your submission, please upload your figure files to the Preflight Analysis and Conversion Engine (PACE) digital diagnostic tool, https://pacev2.apexcovantage.com/. PACE helps ensure that figures meet PLOS requirements. To use PACE, you must first register as a user. Registration is free. Then, login and navigate to the UPLOAD tab, where you will find detailed instructions on how to use the tool. If you encounter any issues or have any questions when using PACE, please email PLOS at . PACE helps ensure that figures meet PLOS requirements. To use PACE, you must first register as a user. Registration is free. Then, login and navigate to the UPLOAD tab, where you will find detailed instructions on how to use the tool. If you encounter any issues or have any questions when using PACE, please email PLOS at figures@plos.org. Please note that Supporting Information files do not need this step.. Please note that Supporting Information files do not need this step.

---

## [Author Response · Author response to Decision Letter 1]

1 Sep 2025

We refer the editor and reviewers to the attached Response to Reviewers letter, which provides point-by-point replies to every comment from the decision letter.

---

## [Decision Letter · Decision Letter 1]

31 Mar 2026

PONE-D-25-30340R1

Variation in Mental Health-Related Sickness Absence: The Role of Occupational Health Professionals

PLOS One

Dear Dr. Timp,

Thank you for submitting your manuscript to PLOS ONE. After careful consideration, we feel that it has merit but does not fully meet PLOS ONE’s publication criteria as it currently stands. Therefore, we invite you to submit a revised version of the manuscript that addresses the points raised during the review process.

The manuscript has been further evaluated by two reviewers, and their comments are available below.

Could you please carefully revise the manuscript to address all comments raised?

If applicable, we recommend that you deposit your laboratory protocols in protocols.io to enhance the reproducibility of your results. Protocols.io assigns your protocol its own identifier (DOI) so that it can be cited independently in the future. For instructions see: https://journals.plos.org/plosone/s/submission-guidelines#loc-laboratory-protocols. Additionally, PLOS ONE offers an option for publishing peer-reviewed Lab Protocol articles, which describe protocols hosted on protocols.io. Read more information on sharing protocols at . Additionally, PLOS ONE offers an option for publishing peer-reviewed Lab Protocol articles, which describe protocols hosted on protocols.io. Read more information on sharing protocols at https://plos.org/protocols?utm_medium=editorial-email&utm_source=authorletters&utm_campaign=protocols..

As the corresponding author, your ORCID iD is verified in the submission system and will appear in the published article. PLOS supports the use of ORCID, and we encourage all coauthors to register for an ORCID iD and use it as well. Please encourage your coauthors to verify their ORCID iD within the submission system before final acceptance, as unverified ORCID iDs will not appear in the published article. *Only* the individual author can complete the verification step; PLOS staff  the individual author can complete the verification step; PLOS staff *cannot* verify ORCID iDs on behalf of authors. verify ORCID iDs on behalf of authors.

We look forward to receiving your revised manuscript.

Kind regards,

Ilse Bloom

Staff Editor

PLOS One

**Journal Requirements:**

Reviewers' comments:

Reviewer's Responses to Questions

**Comments to the Author**

1. If the authors have adequately addressed your comments raised in a previous round of review and you feel that this manuscript is now acceptable for publication, you may indicate that here to bypass the “Comments to the Author” section, enter your conflict of interest statement in the “Confidential to Editor” section, and submit your "Accept" recommendation.

Reviewer #1: All comments have been addressed

Reviewer #2: (No Response)

2. Is the manuscript technically sound, and do the data support the conclusions?

Reviewer #1: Yes

Reviewer #2: Yes

3. Has the statistical analysis been performed appropriately and rigorously?

Reviewer #1: Yes

Reviewer #2: Yes

4. Have the authors made all data underlying the findings in their manuscript fully available?

The PLOS Data policy requires authors to make all data underlying the findings described in their manuscript fully available without restriction, with rare exception (please refer to the Data Availability Statement in the manuscript PDF file). The data should be provided as part of the manuscript or its supporting information, or deposited to a public repository. For example, in addition to summary statistics, the data points behind means, medians and variance measures should be available. If there are restrictions on publicly sharing data—e.g. participant privacy or use of data from a third party—those must be specified. requires authors to make all data underlying the findings described in their manuscript fully available without restriction, with rare exception (please refer to the Data Availability Statement in the manuscript PDF file). The data should be provided as part of the manuscript or its supporting information, or deposited to a public repository. For example, in addition to summary statistics, the data points behind means, medians and variance measures should be available. If there are restrictions on publicly sharing data—e.g. participant privacy or use of data from a third party—those must be specified.

Reviewer #1: Yes

Reviewer #2: No

5. Is the manuscript presented in an intelligible fashion and written in standard English?

Reviewer #1: Yes

Reviewer #2: Yes

6. Review Comments to the Author

Reviewer #1: The authors made substantial improvements in response to the previous feedback. The revisions significantly strengthened the paper through several key changes: the dataset was refined (adjusting the COVID time period and broadening the industry representation), the analysis was enhanced with better controls for organizational level and industry differences, and the focus on occupational health providers was maintained with improved justification and explanation of the interplay between organizational and OHP factors. The paper is now much clearer and more robust than the previous version. I congratulate the authors on the results of their extensive re-working of their paper to address the major criticisms of the reviewers.

There are a few minor issues with the manuscript as reviewed -

Clarification needed: In 3.1 Dataset, there appears to be a contradiction regarding the sickness absence period used in the analysis (2 years versus 3 years). Please clarify which time period was actually applied in the analysis.

Reference formatting: Some references appear inconsistent without mentioning the year (e.g., Salet et al… Barker et al…Di Martin et al. ).

Discussion is missing some references (question marks are in place instead of the reference)

Two minor suggestions for the discussion –

In this sentence “Third, OHPs in occupational health often have more autonomous decision making authority….” Suggest changing “often have” to “may have” as no data were provided to support the assertion that OHP’s have more autonomy than hospital-based physicians. (having practiced in both settings in my country, this was not my experience).

“At the organization level, factors such as HR policies, return-to work procedures, case-management capacity and firm size may play a role.” I would suggest adding “industry sector” to this list.

Reviewer #2: I believe that intentionally delimiting sector variation does indeed reduce generalizability, as noted by reviewer 1. However, the authors had already addressed this in the limitations section of the first draft of the manuscript. In any case, including the effect of sectors in the modeling brought greater realism and allowed for updated estimates of the variation due to occupational health professionals (OHPs) and highlighted the variation due to organizations, allowing for a review of the conclusions and the impact of each source of variation, enabling more informed decision-making. Furthermore, the authors addressed my questions satisfactorily, including the presentation of the cross-classification table, allowing readers to assess the relevance of the modeling adopted.

7. PLOS authors have the option to publish the peer review history of their article (what does this mean?). If published, this will include your full peer review and any attached files.). If published, this will include your full peer review and any attached files.

**Do you want your identity to be public for this peer review?** For information about this choice, including consent withdrawal, please see our  For information about this choice, including consent withdrawal, please see our Privacy Policy..

Reviewer #1: No

Reviewer #2: No

---

## [Author Response · Author response to Decision Letter 2]

7 Apr 2026

We thank the reviewers and editor for their constructive feedback which helped to improve the content, statements, and quality of the paper. Below we discuss how we dealt with these suggestions and hope to have met their expectations with our new version of the paper.

* Responses to questions

4. Have the authors made all data underlying the findings in their manuscript fully available?

The PLOS Data policy requires authors to make all data underlying the findings described in their manuscript fully available without restriction, with rare exception (please refer to the Data Availability Statement in the manuscript PDF file). The data should be provided as part of the manuscript or its supporting information, or deposited to a public repository. For example, in addition to summary statistics, the data points behind means, medians and variance measures should be available. If there are restrictions on publicly sharing data-e.g. participant privacy or use of data from a third party-those must be specified.

Reviewer #1: Yes

Reviewer #2: No

The data contain confidential medical and employment information and cannot be shared publicly under the terms of our ethics approval and data-use agreements. Deidentified individual-level data may be available under controlled access from the Arbo

Unie Data Team (datateam@arbounie.nl) for researchers who obtain ethics approval, sign a data-use agreement, and complete

an approved Data Protection Impact Assessment (DPIA).

* Reaction to reviewer #1

Minor issues:

- Clarification needed: In 3.1 Dataset, there appears to be a contradiction regarding the sickness absence period used in the analysis (2 years versus 3 years). Please clarify which time period was actually applied in the analysis.

/We now explain the time period more clearly in the data section as follows: "Cases are included if the following criteria are met: the employee is between 18 and 65 years old, the sickness absence duration is between one day and three years... The upper limit of three years reflects that some employers extend financial coverage beyond the mandatory two-year period, either because return to work is expected or due to more generous sickness absence regulations. For our analysis, we truncated absences lasting between two and three years to two years, since employers are only legally required to cover the first two years of sickness absence."/

- References: /We revised references and corrected incorrect references./

Two minor suggestions for the discussion -

- In this sentence "Third, OHPs in occupational health often have more autonomous decision making authority..." Suggest changing "often have" to "may have" as no data were provided to support the assertion that OHP's have more autonomy than hospital-based physicians. (having practiced in both settings in my country, this was not my experience). /This is indeed more appropriate, as the statement was based on limited experience rather than empirical data./

- "At the organization level, factors such as HR policies, return-to work procedures, case-management capacity and firm size may play a role." I would suggest adding "industry sector" to this list.

/Changed/

---

## [Editor Report · Decision Letter 2]

12 Apr 2026

Variation in Mental Health-Related Sickness Absence Duration: The Role of Occupational Health Professionals

PONE-D-25-30340R2

Dear Dr. Timp,

We’re pleased to inform you that your manuscript has been judged scientifically suitable for publication and will be formally accepted for publication once it meets all outstanding technical requirements.

An invoice will be generated when your article is formally accepted. Please note, if your institution has a publishing partnership with PLOS and your article meets the relevant criteria, all or part of your publication costs will be covered. Please make sure your user information is up-to-date by logging into Editorial Manager at Editorial Manager® and clicking the ‘Update My Information' link at the top of the page. For questions related to billing, please contact  and clicking the ‘Update My Information' link at the top of the page. For questions related to billing, please contact billing support..

Kind regards,

Marianne Clemence

Staff Editor

PLOS One
---

## [Editor Report · Acceptance letter]

PONE-D-25-30340R2

PLOS One

Dear Dr. Timp,

I'm pleased to inform you that your manuscript has been deemed suitable for publication in PLOS One. Congratulations! Your manuscript is now being handed over to our production team.

Kind regards,

on behalf of

Dr Marianne Clemence

Staff Editor

PLOS One